# Genetic Basis of Grain Size and Weight in Rice, Wheat, and Barley

**DOI:** 10.3390/ijms242316921

**Published:** 2023-11-29

**Authors:** Sebastian Gasparis, Michał Miłosz Miłoszewski

**Affiliations:** Plant Breeding and Acclimatization Institute—National Research Institute in Radzików, 05-870 Błonie, Poland; m.miloszewski@ihar.edu.pl

**Keywords:** grain size, grain weight, genetic regulation, rice, wheat, barley

## Abstract

Grain size is a key component of grain yield in cereals. It is a complex quantitative trait controlled by multiple genes. Grain size is determined via several factors in different plant development stages, beginning with early tillering, spikelet formation, and assimilates accumulation during the pre-anthesis phase, up to grain filling and maturation. Understanding the genetic and molecular mechanisms that control grain size is a prerequisite for improving grain yield potential. The last decade has brought significant progress in genomic studies of grain size control. Several genes underlying grain size and weight were identified and characterized in rice, which is a model plant for cereal crops. A molecular function analysis revealed most genes are involved in different cell signaling pathways, including phytohormone signaling, transcriptional regulation, ubiquitin–proteasome pathway, and other physiological processes. Compared to rice, the genetic background of grain size in other important cereal crops, such as wheat and barley, remains largely unexplored. However, the high level of conservation of genomic structure and sequences between closely related cereal crops should facilitate the identification of functional orthologs in other species. This review provides a comprehensive overview of the genetic and molecular bases of grain size and weight in wheat, barley, and rice, focusing on the latest discoveries in the field. We also present possibly the most updated list of experimentally validated genes that have a strong effect on grain size and discuss their molecular function.

## 1. Introduction

Grain yield is one of the most important agronomic traits in cereal crops. The two main grain yield components are the number of grains per plant or per unit area and grain size, which is positively correlated with grain weight. Grain size was a main factor of cereals domestication and later an artificial breeding target. Grain size can be specified by length, width, thickness, and length/width ratio. Grain length and width are determined by cell number and cell size in the spikelet hulls, while thickness is mostly affected by the grain-filling process [1]. Grain size is a complex quantitative trait determined by multiple genes and factors and has been a research and breeding focus since the beginning of modern agriculture. A major breakthrough in cereal crop yield improvement, known as the green revolution, was achieved in the 1960s by the selection and widespread introduction of crops with beneficial semi-dwarf plant architecture. Later it was discovered that the semi-dwarf phenotype was caused by genes involved in gibberellin signaling, such as semi-dwarf1 (*sd1*) in rice [2], reduced height-1 (*Rht-B1* or *Rht-D1*) in wheat [3], and semi-dwarf1/denso (*sdw1*) in barley [4]. New semi-dwarf varieties bearing these genes produced more grain yield at the expense of reduced straw biomass and were more resistant for lodging (reviewed in [5]).

Rice (*Oryza sativa* L.), wheat (*Triticum aestivum* L.), and barley (*Hordeum vulgare* L.) are the most important small-grain cereal crops. Rice is the second crop (after maize), followed by wheat and barley in terms of global cereal production volume (Faostat, 2021) [6]. Together, they account for more than half of the global production of all cereal crops. Rice and wheat are a primary source of staple food for the human population, and barley is used mostly for animal feed and as raw material for the brewing and distilling industries. Wheat and barley are well-adopted to different environments and are cultivated worldwide, preferentially in the temperate climate zone, whereas rice is cultivated mainly in the southeast regions of Asia. Although the global yield of wheat, barley, and rice substantially increased during recent decades (Figure 1), it resulted mostly from improved agrotechnology and increased use of fertilizers and pesticides, which have a detrimental influence on the environment and must be reduced and alternative eco-friendly strategies must be implemented. Simultaneously, the genetic improvement of grain yield is still insufficient, taking into account the predicted population growth, decrease in arable land, and climate changes. To meet the predicted demand for food, new superior cereal varieties with improved yield potential, nutritional value, disease resistance, and stress tolerance must be developed in the near future. These predictions forced researchers to develop new strategies that would allow further improvements in grain yield beyond the levels achieved with the green revolution genes. One strategy is to seek new superior genes that determine grain size rather than grain number, which is strongly affected by the environment. The trade-off between seed number and seed size is an evolutionarily developed plant adaptation mechanism to variable environments. Seed size is negatively correlated with seed number due to the metabolic cost. If the amount of resources available for reproduction is fixed, then an increase in seed number can only be achieved at the cost of decreased seed weight, and vice versa. However, during evolutionary adaptation, plants established an optimum seed weight that enhances offspring fitness and survival [7,8]. As a consequence, plants show very great plasticity in seed number and a relatively narrow seed size range. In some grasses, even a 100-fold variation in spikelet number was observed in response to different nutrient availability conditions, whereas the seed size variation did not exceed 10% [9]. Grain size in cereal crops results from both evolutionary adaptation and also from agronomic selection, which suggests that this trait is determined by the limited gene pool, and their effect is highly heritable contrary to the grain number. Although the cereal grain size range is relatively limited compared to the grain number range, it still offers the potential to improve yields. The observed grain size ranges from 3 mm to 11 mm in length and 1.2 mm to 3.8 mm in width among domesticated rice varieties, providing areas for substantial improvement in grain yield [10].

The increased availability of high-quality genomic sequences of main crop species and the application of new genome editing techniques based on CRISPR/Cas9 technology [11,12] resulted in considerable progress in the genetic background of cereal grain yield. The genetic and molecular bases of grain size have been relatively well explored in rice during the past decade, which allowed for the identification and characterization of numerous genes that determine this trait. Many genes reviewed in this paper were evaluated in field trials and appeared to have a strong, heritable effect on grain size regardless of environmental conditions. Compared to rice, the genetic background of grain size in wheat, barley, and other cereal crops remains largely unexplored and the number of characterized genes is considerably less. This difference may be explained by the fact that rice is considered as a model plant for other cereal crops, mostly because of its relatively small genome (approximately 460 Mb) consisting of 12 chromosomes (2n = 24). For comparison, the barley genome (2n = 14) is approximately 4900 Mb and the largest genome among cereal species of hexaploid wheat (2n = 6X = 42) is approximately 1700 Mb [13]. Despite differences in chromosome number and size, the genomes of cereal species show a relatively high conservation level. For instance, the comparison between annotated rice genes and mapped wheat genes revealed 13 blocks of collinearity, which cover 90% and 83% of rice and wheat genomes, respectively [14]. This high degree of conservation of cereal genome structure offers an opportunity to utilize the knowledge gained in rice to other important crops, such as wheat and barley, using comparative genomics and homology-based cloning approaches. 

In this review, we present a comprehensive overview of genetic and molecular determinants of grain size and weight in wheat, rice, and barley. Moreover, we compiled an updated list of experimentally validated genes with accession numbers, which determine grain size and weight in rice, wheat, and barley (Appendix A). The function of these genes is discussed in the context of molecular mechanisms and regulatory pathways in which they are involved. 

## 2. Factors Determining Grain Size and Weight during Plant Development

Final grain yield in cereals is determined by the interaction of genotype with environment. Depending on the plant’s developmental stage, the determinants of grain size and weight can be divided into pre-anthesis and post-anthesis factors (Figure 2). The events that occur during the pre-anthesis phase have an indirect influence on grain size by modulating the so-called source-to-sink relationship, where the source is the capacity of assimilates determined by leaf area and sink means grain number per m^2^. The first critical step of the pre-anthesis phase occurs during tillering and spikelet formation. The strong competition for assimilates between developing tillers and spikelets determines the tiller survival and number of fertile spikelets per spike and, consequently, grain number per plant. This process is strongly affected by environmental factors, such as temperature, photoperiod, and water availability; however, genetic factors that regulate plant architecture, vernalization, or flowering time may also contribute to grain number determination. For instance, in semi-dwarf phenotypes, the competition for assimilates is less strong from the stem, which enhances the supply to reproductive tissues and improves spikelet fertility. Spike architecture and the number of spikelets per spike are also determined by genotype. An example is the *Vrs1* and *INT-C* genes in barley that determine whether two-rowed or six-rowed spikes are generated [15,16]. The duration of the pre-anthesis phase and flowering time are also key factors that determine grain number. Longer spikelet initiation phase and delayed flowering time positively affect spikelet fertility and seed setting (reviewed in [17]). A prolonged pre-anthesis phase may also enhance the accumulation of assimilates that will be available during grain development. However, it exposes the plant to a greater risk of adverse environmental conditions that may occur during anthesis and early grain development stages. Other critical steps of the pre-anthesis phase, which may affect grain number, are microsporogenesis and pollen development, which are particularly sensitive to abiotic stress factors. Even a short drought incidence during microspore mother cell meiosis perturbs the subsequent microsporogenesis, resulting in sterile pollen and decreased seed setting rate [18]. The pre-anthesis factors that determine the final grain number have an indirect influence on grain size and weight since these traits are usually negatively correlated with the grain number per plant [9]. Thus, the relationship between grain number and grain size is an important factor that contributes to the final grain yield.

Grain size and shape can also be determined during the pre-anthesis phase by floret architecture. A hull consisting of lemma and palea is a floral organ that surrounds the developing grain after pollination. Its size is strongly determined by genes regulating the cell division or cell expansion in the early inflorescence development stages. In turn, the grain length, width, and weight are strictly associated with the hull size. 

The post-anthesis phase starts after pollination and continues until grain maturity. Post-anthesis factors can determine grain size both in a direct or indirect way. 

Grain development can be divided into the following three main phases: cell proliferation, grain filling, and grain desiccation. The cell proliferation phase starts just after fertilization and completes at approximately 20 DAP (days after pollination), and further endosperm growth occurs by cell expansion [19]. The endosperm cell number is positively correlated with the grain weight, thus the genes controlling cell division and cell expansion during endosperm development are key factors that have a direct influence on this trait. Endosperm development is initiated by the mitotic divisions of syncytium, a single multinucleate cell, and is followed by its cellularization and differentiation of the endosperm cells. This is a critical step for the subsequent endosperm growth, and a disruption of this process leads to the formation of unfilled grains (see Section 3.7). The grain-filling phase starts approx. between 5 and 10 DAP [20] and is initiated by formation of large (A-type) and small (B-type) starch granules. Grain filling depends on the assimilate supply both from current photosynthesis and remobilization of reserves from vegetative tissues. The availability of assimilates, which depends on leaf area, can be, to some extent, determined by genetic factors during the post-anthesis phase. For instance, the genetic regulation of ethylene signaling may delay leaf senescence, resulting in an increased production of assimilates during the grain-filling stage and, consequently, an increased grain yield (see Section 3.4.5). Apart from starch, storage proteins are also synthesized and accumulated during the grain-filling phase. Prolamins, glutelins, globulins, and albumins that represent main fractions of cereals’ storage proteins are synthesized in the secretory pathway and stored in protein bodies, derived from the vacuole or endoplasmic reticulum lumen [21]. The duration of grain filling depends mostly on environmental conditions, whereas the grain-filling rate, which relates to the rate of dry matter accumulation, is strongly determined by the genotype. The genes controlling starch synthesis, storage protein synthesis, and sugar transporter activity are important during this process. The grain-filling phase completes at physiological maturity, at which caryopsis reaches its greatest size and weight, which is later gradually reduced during desiccation. 

## 3. Genetic and Molecular Regulation of Grain Size

### 3.1. G Protein Signaling

G protein signaling is a conserved mechanism in animal and plant cells for transducing extracellular signals to the downstream intracellular effectors. Heteromeric G protein complex consists of three subunits bound to the cell membrane, Gα, Gβ, and Gγ. The complex is activated by binding an extracellular ligand to the trans-membrane G protein-coupled receptor (GPCR), which acts as a guanine-nucleotide exchange factor. GPCR induces the exchange of GDP to GTP in the Gα subunit, leading to the disassociation of the inactive G protein complex into active Gα-GTP monomers and Gβγ dimers. Both subunits interact independently with an array of downstream effectors to activate different signaling cascades.

Compared to the animals, where dozens of G protein variants were identified, plants have a very reduced number of canonical G protein components, i.e., one Gα, one Gβ, and seven Gγ proteins. Apart from these classes, three plant-specific extra-large G proteins (XLGs) were classified as non-canonical Gα subunits. Also, the plant G protein complex self-activates without any interaction with GPCRs. Despite the limited number of G protein variants, G protein signaling in plants is involved in the regulation of a broad range of developmental and physiological processes, including plant architecture, nitrogen and water use efficiency, nitrogen fixation, organ development, chloroplast development, hormonal regulation, and pathogen responses and abiotic stress responses, such as oxidative stress and ROS signaling (reviewed in [22,23]). Importantly, G protein signaling also controls grain size in cereals (Table 1). Studies in rice indicated that components of all three G protein subunits are involved in grain size regulation. The Gα and Gβ subunit genes *RGA1* and *RGB1* are both positive regulators of cell proliferation and can act independently from each other. Mutation in either *RGA1* or *RGB1* led to shorter grains [24,25]. However, the effect of RGB1 on grain size depends also on the interaction with other Gγ components, RGG1, RGG2, GS3, and DEP1. For instance, the overexpression of two canonical Gγ subunits, RGG1 and RGG2, both interacting with the RGB1, led to decreased grain size, indicating that both proteins negatively affect grain development, either by suppressing cell proliferation in lemma (RGG1) or cell elongation (RGG2) [26,27]. A more complex grain size regulation mechanism was observed in the case of three non-canonical Gγ subunits, GS3 (GRAIN SIZE3), DEP1 (DENSE AND ERECTED PANICLE1), and GGC2. *GS3* is a major QTL (Quantitative Trait Locus) for grain length and weight first characterized in rice [28]. Plants with the *GS3* wild-type allele produce medium-sized grains, and the other forms, short or long grains, result from mutations in different functional domains. For example, a loss-of-function mutation within the OSR domain (organ size regulation) in the *GS3-3* allele results in longer grains, while the deletion of the C-terminal Cys-rich region in allele *GS3-4* results in super short grains [29,30]. DEP1 and GGC2 are other Gγ subunits that positively regulate grain size in rice. DEP1 and GGC2 can act individually or in combination, and their impact on grain size is additive [31]. DEP1 is encoded by a major QTL for rice panicle architecture, and its N-terminal and C-terminal domains show similarity to GS3 [32,33]. Similar to *GS3*, a mutation in the OSR domain in DEP1 decreases grain size, but simultaneously, it also affects the panicle architecture, leading to more dense and erected panicles and, consequently, to increased grain number per plant [32]. Gβγ complex consisting of RGB1 and DEP1 or GS3 is involved in the signal transduction to the downstream effector OsMADS1. This is a MADS-domain transcription factor, and its alternatively transcribed truncated form OsMADS^lgy3^ can interact directly with GS3 or DEP1. Interaction with GS3 or DEP1 enhances *OsMADS1* transcriptional activity, which is associated with the formation of longer and slender grains and increased grain yield [34].

G protein signaling genes were also identified and characterized in wheat and barley. Wheat has a single gene for the Gα and Gβ subunits, and four genes for the Gγ subunits. Each gene has three homeologous copies on the A, B, and D genomes [35,36]. Although the effect of Gα and Gβ proteins on grain yield has not yet been determined in wheat, it should be expected that both subunits have a homologous function to their rice counterparts due to great sequence similarity. Such functional homology has been observed in the case of wheat *GS3*, which negatively regulates grain length and weight [37,38]. In turn, the effect of *DEP1* on plant architecture is even stronger in wheat than in rice, as its knockout results in an extreme dwarf phenotype [39]. In barley, the effect of Gα and one of the Gγ subunits on grain size and yield were evaluated. A mutation in *Brh1*, which encodes the Gα subunit, is associated with a semi-dwarf phenotype and decreased grain yield [40]. The *dep1* mutants in barley exhibit a decreased grain size similarly to rice; however, the decreased grain number per plant is an opposite effect [41]. 

**Table 1 ijms-24-16921-t001:** List of genes controlling grain size in rice, wheat, and barley involved in G protein signaling.

Gene or QTL Name	Gene ID ^1^	Protein Category	Type of Regulation	Regulated Process	Affected Trait ^2^	References
**Rice**						
Rice G Protein Alpha Subunit (RGA1)	Os05g0333200	G protein α subunit	positive	cell proliferation	grain size	[24]
Rice G Protein Beta Subunit (RGB1)	Os03g0669100	G protein β subunit	positive	cell proliferation	grain size	[25]
Grain Size 3 (GS3)	Os03g0407400	G protein γ subunit	negative	cell proliferation	grain length	[28,29]
Dense And Erect Panicle 1 (DEP1)	Os09g0441900	G protein γ subunit	positive	cell proliferation	grain length	[32,33]
GGC2	Os08g0456600	G protein γ subunit	positive	cell proliferation	grain length	[31]
RGG1	Os03g0635100	Type-B G protein γ subunit	negative	cell proliferation	grain size	[27]
RGG2	Os02g0137800	Type-A G protein γ subunit	negative	cell proliferation	grain size	[26]
**Wheat**						
TaGS3	TraesCS4A02G474000	G protein γ subunit	negative	cell proliferation	grain size	[37,38]
TaDEP1	FJ039902	G protein γ subunit	positive	cell proliferation	grain size	[39]
**Barley**						
Brachytic1 (Brh1)	AF267485	G protein α subunit	positive	cell proliferation	grain size	[40]
HvDEP1	FJ039903	G protein γ subunit	positive	cell proliferation	grain size	[41]

^1^ Gene IDs are according to Release 57 of EnsemblPlants database [42], otherwise accession numbers to GeneBank database [43] of NCBI are provided; ^2^ Grain size is an affected trait when both grain length and width were changed or no specific parameter except of grain size was provided in the source publication.

### 3.2. Ubiquitin–Proteasome System

The ubiquitin–proteasome system (UPS) is a crucial mechanism of selective protein degradation in eucaryotic cells. The major components of UPS include ubiquitin, ubiquitin-activating enzyme (E1), ubiquitin-conjugating enzyme (E2), ubiquitin-protein ligase (E3), 26S proteasome, and target protein. The ubiquitination process is initiated by a covalent attachment of ubiquitin to the target protein through the conjugation cascade mediated by the E1, E2, and E3 enzymes. The ubiquitin-target protein conjugates are then recognized and degraded by the 26S proteasome [44]. The UPS can rapidly inactivate the abnormal or regulatory proteins, providing an efficient and selective regulatory mechanism involved in many physiological processes [45,46]. Grain size in cereals is also controlled by the genes involved in the UPS. Most of these genes encode different types of E3 enzyme from the ubiquitin conjugation pathway, and their positive or negative influence on grain size is mediated, in most cases, by enhancing or inhibiting cell proliferation (Table 2).

*GW2* was the first QTL cloned in rice which appeared to be involved in the ubiquitin–proteasome pathway. *GW2* encodes a RING-type E3 ubiquitin ligase. Deletion in the *GW2* coding region, which resulted in a premature stop codon, caused increased grain width and weight [47]. A similar effect was observed in wheat carrying *GW2* mutant alleles [48,49]. In rice, the negative grain size regulation by *GW2* is associated with the ubiquitin-mediated degradation of a downstream substrate, CC-type glutaredoxin (OsGRX8) encoded by *WG1*. *WG1* interacts with and suppresses the *OsbZIP47* transcription factor, which is a negative regulator of cell proliferation [50]. Consequently, *GW2* loss of function promotes cell proliferation in hulls and increases the grain-filling rate, leading to wider and heavier grains. In wheat, *GW2* interacts with the ubiquitin receptor encoded by *TaDA1*. *TaDA1* silencing resulted in increased grain size and weight, which suggests that *DA1* has an additive, negative effect with *GW2* for grain size and TGW (thousand grains weight) [51]. Other types of E3 ubiquitin ligases are also involved in grain size regulation in even more complex mechanisms through interaction with the G protein signaling. An example is U-Box E3 ubiquitin ligase in rice encoded by *TUD1*, which interacts with the Gα subunit encoded by *RGA1*. *TUD1* functions in the brassinosteroid signaling pathway and the *tud1* mutants produce extremally short grains because of inhibited cell proliferation [52]. Another E3 ubiquitin ligase encoded by *CLG1* targets the Gγ subunit GS3, which is a negative grain size regulator. Wild-type *CLG1* overexpression in rice resulted in larger grains, while the mutant alleles gave the opposite result [53].

Recently, in wheat two additional RING-type E3 ubiquitin ligases were identified after GW2. They are TaSDIR1-4a and ZnF-B, both negatively affecting grain weight. The TaSDIR1-4A mode of action is not known; however, Wang et al. [54] demonstrated that its expression might be repressed by TaERF3 (ethylene response transcription factor), which is attributed to the increased TGW. Complete deletion of *ZnF-B*, which acts as a brassinosteroid (BR) signaling activator, resulted in a semi-dwarf phenotype with larger grains and an increased grain yield up to 15.2% [55]. This new semi-dwarf form is advantageous to the previously developed semi-dwarf varieties that rely on the *Rht-B1b* and *Rht-D1b* alleles as they produce smaller grains and require greater nitrogen fertilization to compensate for the reduced yield. 

Ubiquitination can be reversed by deubiquitinating enzymes (DUBs) with the protease activity that can cleave off the ubiquitin from the target protein [56]. Grain size can also be controlled by this mechanism. Contrary to the ubiquitination pathway, only two deubiquitinating enzyme genes with direct impact on grain size were identified in rice and wheat, *WTG1* and *LG1*. OsOTUB1/WTG1 (WIDE AND THICK GRAIN1), which is a functional otubain protease in rice, promotes cell elongation in epidermal cells in the spikelet hulls, causing long and thin grain formation. The *wtg1* loss of function mutants produce wide, thick, and heavier grains [57]. Moreover, the mutation has positive effects on plant architecture, characterized by larger panicles, stronger culms, and a greater grain number per panicle [58]. In wheat, however, a reduced OTUB1 expression led to decreased tiller number [59]. 

*LG1* (LARGE GRAIN1) encodes another deubiquitinating enzyme, ubiquitin-specific protease15 (OsUBP15). The dominant *LG1* (*lg1-D*) mutation was associated with a 30.8% increase in seed width and a 34.5% increase in 1000-grain weight as compared to the wild-type [60]. OsPUB15 interacts with the ubiquitin receptor OsDA1, similar to GW2 in wheat, which may suggest that OsPUB15 and GW2 are mutually dependent on grain width regulation. 

So far, there are no reports of the UPS genes associated with barley grain size; however, one can expect that the barley orthologs should have the same function as their rice and wheat counterparts, due to the strong conservation of crucial ubiquitination genes and the close relationship between these species. 

**Table 2 ijms-24-16921-t002:** List of genes controlling grain size in rice and wheat involved in the ubiquitin–proteasome system.

Gene or QTL Name	Gene ID	Protein Category	Type of Regulation	Regulated Process	Affected Trait	References
**Rice**						
Grain Width 2 (GW2)	Os02g0244100	RING-type E3 ubiquitin ligase	negative	cell proliferation	grain width	[47,50]
U-Box E3 Ubiquitin Ligase (TUD1)	Os03g0232600	U-box domain-containing protein 75	positive	cell proliferation	grain size	[52]
Chang Li Geng 1 (CLG1)	Os05g47780	RING E3 ligase	positive	cell proliferation	grain length	[53]
Wide And Thick Grain 1 (WTG1)/OsOTUB1	Os08g0537800	Otubain-like protease with deubiquitination activity	negative	cell expansion	grain length	[57,58]
Large Grain 1 (LG1)	Os02g0244300	Ubiquitin-specific protease 15	positive	cell proliferation	grain width	[60]
**Wheat**						
Grain Width 2 (GW2)	TraesCS6A02G189300	RING-type E3 ubiquitin ligase	negative	cell proliferation	grain width	[48,49]
TaDA1	TraesCSU01G007800	ubiquitin receptor DA1	negative	cell proliferation	grain size	[51]
TaSDIR1-4A	TraesCS4A02G096000	RING finger E3 ligase	negative	ns ^1^	grain size	[54]
ZnF-B	TraesCS4B02G042900	zinc-finger RING-type E3 ligase	positive	cell proliferation	grain size	[55]
Wide And Thick Grain 1 (WTG1)	TraesCS7A02G263900	Otubain-like protease with deubiquitination activity	negative	cell expansion	grain length	[59]

^1^ Not specified.

### 3.3. MAPK Signaling

Mitogen-activated protein kinase (MAPK) cascades are highly conserved signaling modules I eucaryotes that transduce extracellular signals downstream of receptor-like protein kinases. Each cascade module is composed of three protein kinases that act sequentially. The kinase at the cascade top is MAPKKK (or MEKK), which receives signals from receptors and initiates phosphorylation of downstream kinases, MKKKs (MAPK kinase kinases), and MAPKs (MAPK kinases). MAPK signaling plays an essential role in plant immunity and stress response, as well as in the regulation of growth and development [61,62]. MAPK signaling is also involved in grain size regulation in cereals (Table 3). In rice, the main cascade module, which regulates grain size by enhancing cell proliferation, consists of OsMKKK10, OsMKK4, and OsMAPK6, which are encoded by *SMG2*, *SMG1*, and *DSG1*, respectively. The loss of function mutants *smg2*, *smg1*, and *dsg1* produces smaller grains as compared to the wild-type [63,64,65,66]. The sequential activation of OsMKKK10, OsMKK4, and OsMAPK6 affects the BR signaling pathways, which results in increased cell proliferation and cell expansion in spike hulls. The crosstalk between the MAPK cascade and BR signaling is mediated by OsWRKY53, which acts as a downstream receptor of the MAPK module [67]. The activity of the OsMKKK10-OsMKK4-OsMAPK6 cascade can be modulated by other genes, which has consequences in the regulation of grain size by these modules. 

*GSN1* in rice encodes a MAPK phosphatase OsMPK1, which has a negative effect on grain size by OsMAPK6 dephosphorylation and its inactivation [68]. Simultaneously, *GSN1* positively regulates the grain number per panicle. 

OsRac1, a Rho-family GTPase has an opposite effect to GSN1, as it positively regulates grain size by OsMPAK6 phosphorylation. OsRac1 overexpression resulted in an increased hull cell number and grain-filling rate, which consequently resulted in larger grains [69].

Contrary to rice MAPK kinases, wheat TaMPK3 is a negative regulator of grain weight; however, the regulation mechanism is still elusive. Plants overexpressing TaMPK3 formed narrower grains with a decreased TGW [70]. These changes might be associated with a decreased ABA sensitivity and a drought tolerance observed in the tested plants.

**Table 3 ijms-24-16921-t003:** List of genes controlling grain size in rice and wheat involved in MAPK signaling.

Gene or QTL Name	Gene ID	Protein Category	Type of Regulation	Regulated Process	Affected Trait	References
**Rice**						
Small Grain 1 (SMG1)	Os02g0787300	Mitogen-activated protein kinase kinase 4 (MKK4)	positive	cell proliferation	grain size	[63]
Small Grain 2 (SMG2)	Os04g0559800	Mitogen-activated protein kinase kinase 10 (MKK10)	positive	cell proliferation	grain size	[66]
WRKY-type transcription factor WRKY53	Os05g0343400	Transcription factor WRKY53	positive	cell proliferation	grain size	[67]
Dwarf and Small Grain1 (DSG1)/OsMAPK6	Os06g0154500	Mitogen-activated protein kinase 6	positive	cell proliferation	grain size	[64,65]
Grain Size and Number 1 (GSN1)/LARGE8	Os05g0115800	Mitogen-activated protein kinase phosphatase OsMKP1	negative	cell proliferation	grain size	[68]
ROP GTPase (OsRac1)	Os01g0229400	Rho-family GTPase	positive	cell proliferation	grain size	[69]
**Wheat**						
TaMPK3	TraesCS4D02G198600	Mitogen-activated protein kinase	negative	ns ^1^	grain width	[70]

^1^ Not specified.

### 3.4. Phytohormone Signaling

#### 3.4.1. Auxin

Auxin is an essential phytohormone required for all major developmental processes, such as embryogenesis, endosperm development, root development, seed development, seedling growth, vascular patterning, phyllotaxis, flower development, and de novo organogenesis (reviewed in [71]). Several major genes related to auxin metabolism or signaling are grain size regulators. The grain size may be affected by genes controlling auxin synthesis (*OsYUC11*, *TSG1/FIB*), local activation (*TGW6*), auxin transport (*BG1*), and signaling (*qTGW3*, *Gpn4/LAX2*, *OsIAA3*, *OsARF4*, *OsARF6*, *OsARF12*, *OsAUX3*) (Table 4).

IAA is synthesized in the TAA-YUC pathway (tryptophan aminotransferase of Arabidopsis—YUCCA flavin monooxygenase). *TSG1* encodes a tryptophan aminotransferase, which is involved in local IAA biosynthesis. *tsg1* mutant plants with significantly reduced auxin levels produced more tillers, but with a reduced grain size and number [72]. *OsYUC11*, a flavin-containing monooxygenase encoding gene, is another key gene from the auxin biosynthesis pathway that has a positive effect on grain size. Auxin deficient plants resulting from a *OsYUC11* mutation produced smaller grains due to suppressed endosperm development [73]. Normal grain development was recovered by exogenous auxin application, which suggests that the local auxin biosynthesis by *OsYUC11* is organ specific.

*THOUSAND-GRAIN WEIGHT 6* (*TGW6*), one of the major grain weight QTLs, was first identified in the Indian landrace rice Kasalath [74]. *TGW6* encodes a hydrolase, which hydrolyses IAA-glucose, releasing free IAA. *TGW6* may regulate the timing of the transition from the syncytial to the cellularization phase during early endosperm development, which depends on the IAA level. In *tgw6* loss of function mutants, the IAA level in grains at 3 DAF (days after flowering) was significantly less as compared to wild-type, which promoted cell proliferation and increased endosperm length. Moreover, *tgw6* plants showed increased carbohydrate accumulation in the source organs before heading, which later supplied the developing grains during starch synthesis. *TGW6* orthologs were also identified in wheat, and certain allelic variants were associated with increased grain size and TGW [75,76]. These data indicate that *TGW6* is a negative regulator of grain length and weight by limiting endosperm development rather than cell expansion or cell proliferation in husks. This distinct regulatory mechanism discerns *TGW6* from most grain size-related genes. Grain size may also be regulated by auxin transport and response. 

BIG GRAIN 1 (BG1) was recently identified as a cytoplasmic membrane localized protein, which is involved in basipetal auxin transport. *BG1* overexpression enhances cell proliferation and cell expansion in hulls and results in larger grains. In turn, *BG1* silencing gives the opposite effect and also reduces auxin transport and the sensitivity to auxin, which confirms the primary role of *BG1* in auxin transport and response [77,78]. In wheat, *BG1* overexpression increased grain size; however, the final grain yield did not change due to the reduced grain number per plant. Interestingly, *BG1* overexpression also altered the grain nutritional profile by reducing the Zn level and protein content [79].

Auxin reception and signaling are also important in grain size regulation. The AUX/IAA (auxin receptors/transcriptional repressors) and ARF (auxin response factors) genes are key regulation components. A well-described example is the *miR167-OsARF6-OsAUX3* module, which regulates grain length and weight in rice. Qiao et al. [80] demonstrated that OsARF6 is an upstream transcription factor for AUX3, and both elements have negative impacts on grain length and weight. Conversely, a positive effect was observed when *OsARF6* was silenced by *miR176a*. Interestingly, the opposite regulation was observed after OsARF12 silencing by a *miR167* member, which suggests that this transcription factor is a positive grain size regulator [81]. 

In the OsGSK5-OsARF4 module, OsARF4 is phosphorylated by OsGSK5 (glycogen synthase kinase). The suppression of either OsARF4 or OsGSK5 enhances grain size [82].

Another complex signaling pathway includes the interaction between four genes. Zhang et al. [83] proposed that Gnp4/LAX2 positively regulates grain length and weight by interaction with the OsIAA3-OsARF25-SMOS1 module. In this module OsARF25 binds to the *SMOS1* promoter, an organ size regulator [84], and promotes its transcription, which has a positive effect on grain length and TGW. 

#### 3.4.2. Cytokinins

Cytokinins are a group of phytohormones that stimulate cell division, and their primary function is regulation of organ formation. They are also involved in seed setting and development, thus the genes related to cytokinin metabolism and signaling may be important in grain size regulation [85,86] (Table 4).

*BIG GRAIN 3* encodes a purine permease OsPUP4, which regulates cytokinin transport and distribution [87]. A dominant mutant of *BG3* (*bg3-D*), as well as plants overexpressing *BG3*, produce remarkably larger grains caused by the elevated cytokinin level in panicles, which enhances both the hull cell proliferation and cell expansion [87]. Moreover, *BG3* can be activated by AGO2 (ARGONAUTE2), and its overexpression increases the grain size at a level similar to BG3 [88]. A similar positive influence on grain size was shown by another purine permease homolog, *OsPUP7* [87,89]. *OsPUP7* is, in turn, regulated by *OsPIL15,* encoding a basic helix-loop transcription factor, which binds to the *OsPUP7* promoter region. In the *OsPIL15* knockout plants, the active cytokinin level increased in the panicles, which indicates that *OsPIL15* is a negative regulator of cytokinin transport [90].

*OsSGL* (*STRESS_tolerance* and *GRAIN_LENGTH*) is an abiotic stress related gene in rice that enhances grain length and yield [91]. *OsSGL* encodes a putative DUF1645 protein family member of an unknown function. *OsSGL* overexpression increased grain length, weight, and grain number per panicle. Transcriptomic analyses showed that *OsSGL* overexpression was associated with the up- or downregulation of many genes involved in stress-response, cell cycle, and cytokinin signaling. The enhanced cell division in grain husk and the grain-filling rate suggest that *OsSGL* regulates meristematic activity via cytokinin signal transduction.

*OsPPKL1* encoding a protein phosphatase with a kelch-like domain is an inhibitor of cytokinin phosphorelay in rice. A mutation at a specific PPKL1 amino acid activates the cytokinin response leading to an increased grain size [92]. Interestingly, *OsPPKL1* is also known as a brassinosteroid signaling component (see Section 3.4.4); however, Liu et al. [92] demonstrated that its cytokinin inhibiting function is independent of the phosphatase domain that is required in brassinosteroid signaling. These data suggest that *OsPPKL1* is a key regulator of plant hormone signaling, which provides a balance between cytokinis and brassinosteroids in the regulation of grain development.

*CKX* gene family members encode oxidase/dehydrogenases, which are responsible for an irreversible inactivation of the bioactive form of cytokinins by cleaving the N6-side chain [85]. The downregulation of *CKX* members may increase grain yield in rice, wheat, and barley. In rice, *OsCKX2* silencing increased the tiller number, grain number per plant, and TGW up to 15% [93]. In wheat, the silencing of *CKX* orthologs *CKX2.2.2* and *CKX2.2.1* significantly increased TGW, although the grain number was reduced by 24% [94]. Another wheat *CKX* member, *TaCKX6,* is also involved in grain yield regulation. The allelic variant designated TaCKX6a02 is associated with a significant increase in grain size, TGW, and grain-filling rate [95,96].

*CKX* genes also control grain yield in barley, however, the influence of individual *HvCKX* genes on grain size and weight remains unclear. RNAi-based silencing of a few *HvCKX* members, including *HvCKX1,* resulted in an increased grain size and weight [97,98]. However, the knockout of a single *HvCKX1* or *HvCKX3* gene led to a slightly increased grain number, but TGW did not change. This contradiction result may be because the complete knockout of *HvCKX1* or *HvCKX3* induces a homeostatic feedback loop, which alters the expression of other *CKX* members or genes involved in cytokinin biosynthesis and activation, such as *IPT* or *LOG* [99,100].

#### 3.4.3. Gibberellins

Gibberellin (GA) is a phytohormone, which regulates essential processes during plant development, such as seed germination, organ growth, transition from vegetative to reproductive state, and flower, fruit, and seed development [101]. GA manipulation has been utilized in agriculture and has allowed for the introduction of dwarfing alleles, which greatly increased the yield of wheat and rice in the achievement known as the green revolution. Recent studies showed that newly identified genes related to GA metabolism and signaling can be utilized for further grain yield improvement in cereals (Table 4).

The *GDD1* (*Gibberellin-Deficient Dwarf 1*) gene in rice encodes the kinesin-like protein BRITTLE CULM12 (BC12), which possesses transactivation activity and binds to the *ent*-kaurene oxidase (*KO2*) promoter region, which encodes an enzyme involved in the GA biosynthesis pathway. The *dgg1* mutants exhibit a greatly reduced grain length, which is associated with the decreased GA level and, consequently, inhibited cell elongation [102]. Thus, *GDD1/BC12* is one of the key grain length regulators via affecting GA-mediated cell expansion.

*GW6* is a rice QTL that controls grain size through GA regulation. GW6 encodes a GA-regulated GAST family protein that is greatly expressed in young panicles. GW6 is induced by GA and its overexpression has positive effects on grain width and weight by promoting cell expansion in the spikelet hull while the knockout gives opposite results [103]. Moreover, the *GW6* expression level is associated with a natural variation in the *cis* element CAAT-box in its promoter. *OsGASR9* is another GAST family member that positively regulates grain size and yield by a similar, GA-stimulated mechanism as *GW6* [104].

GA is also regulated by numerous transcription factors, and two of them have a direct impact on grain size and yield. The first is OsMADS56, encoded by *GL10*, which is a positive regulator of grain length and weight. *GL10* promotes longitudinal cell growth in the glumes. Plants with the natural allelic variant *gl10* containing a 1 kb deletion in the first exon exhibit delayed flowering, shorter grains, and reduced grain weight [105]. The second transcription factor is OsWRKY36, which in turn negatively regulates grain size and weight. OsWRKY36 acts as a GA signaling repressor by binding directly to the *SLR1* promoter, a GA signaling DELLA-like inhibitor, and enhances its transcription [106]. The DELLA protein-mediated suppression of GA signaling was also observed in wheat. Wheat cultivars with the semi-dwarf alleles *Rth-B1b* and *Rth-D1b* accumulate high levels of DELLA proteins that attenuate GA-induced plant growth, which is also exhibited in the smaller grains. Plants with deleted *Rth-B1b* and *Rth-D1b* alleles produced significantly larger grains, which increased the grain yield by up to 15% [49].

#### 3.4.4. Brassinosteroids

Brassinosteroids (BRs) are a class of plant steroid hormones that control many important traits, such as plant architecture, flowering time, seed yield, and stress tolerance [107]. To date, some BR-related genes that control grain size and weight are the most numerous among other plant hormones (Table 4).

BRs are necessary for proper seed development as they promote cell expansion in hulls. An inactivation of genes that control BRs biosynthesis in rice, such as *D2* [108], *D11* [109,110,111], *BRD2* [112], and *SLG* [113], leads to a dwarf phenotype with short grains. These genes encode key BR biosynthesis enzymes, cytochrome P450 family members (*D2* and *D11*), delta (24)-sterol reductase (*BRD2*), and BAHD acyltransferase-like protein (*SLG*). *TaD11* is a *D11* wheat ortholog, and similarly to rice, its loss of function mutant *tad11-2a-1* displayed dwarfism and smaller grains [114]. 

BR signaling is controlled by numerous genes that are involved in different molecular pathways, including BR response, BR homeostasis, and transcriptional regulation.

*BRI1*, *BAK1*, *GSK2*, and *BZR1* are key BR signal transduction genes. BRASSINOSTEROID INSENSITIVE1 (BRI1) is a plasma membrane localized leucin-rich-repeat receptor-like kinase (LRR-RLK), and functions as a BR receptor. The loss of function of *BRI1* results in a dwarf phenotype with smaller grains and a significantly reduced grain yield [115]. After BR reception, BRI1 interacts with the LRR-RLK protein BAK1 (BRASSINOSTEROID INSENSITIVE 1-associated receptor kinase 1), which also positively regulates grain size. A rice small grain mutant (*sg2*) possesses a *OsBAK1* mutation, which results in decreased plant height, grain size, and grain number [116]. *XIAO* encodes another LRR-RLK protein that positively influences grain length; thus, it can be concluded that it plays a similar role together with BRI and OSBAK1 at the top of the BR signaling cascade. 

Another component in BR signal transduction is glycogen synthase kinase-3 (Shaggy-like Kinase 2 (GSK2)), which is a negative BR signaling regulator. Plants overexpressing *GSK2* exhibit typical symptoms for BR-deficient mutants, including small grains [117]. *GSK2* downregulates downstream transcription factors, BRASSINAZOLE RESISTANT1 (BZR1) and two GRAS family members, DWARF AND LOW-TILLERING (DLT and DLT2) [110,117], among others. BZR1 is a positive grain size regulator. *OsBZR1* encodes a transcriptional activator, which directly binds to BR-responsive elements in the upregulated gene promoters, such as *CSA* (*Carbon Starved Anther*) or *DGS1* (*Decreased Grain Size 1*). *CSA* enhances sugar accumulation in anthers and developing seeds [110], and *DGS1* promotes cell expansion in grain hulls [118]. Intriguingly, both *DLT* and *DLT2* are positive BR signaling regulators since the *dlt* and *dlt2* mutants exhibit a semi-dwarf phenotype in BR-deficient plants; however, the mutation has positive impacts on grain width and TGW, which is in contrast with other BR-related genes causing dwarfism [119,120]. Cytological and biochemical studies revealed that DLT, DLT2, and BZR1 can interact with each other in a transcriptional complex that mediates BR signaling. Zou et al. [120] suggested that DLT2 probably modulates the transcriptional activity of DLT and BZR1; however, the mechanism of this modulation remains unclear.

SG2 (Small Grain2) is another target phosphorylated by GSK2 in the BR-response pathway. *SG2* encodes a ribonuclease H-like domain protein, which is a positive BR signaling regulator and has a positive influence on grain size by enhancing cell proliferation [121]. Moreover, SG2 may interact with an *OVATE* family transcriptional repressor, OsOFP19, which plays an antagonistic role to SG2 in grain size regulation.

In turn, *OsOFP8* and *OsOFP14* suppress *GS9* transcriptional activity, which encodes a putative transcriptional activator and negatively regulates grain length [122,123]. The transcription co-regulation involving *OsOFP8*, *OsOFP14*, and *GS9* can be further modulated by *GSK2* [122]. Other *OVATE* family members, *OsOFP1* [124] and *OsOFP3* [125], negatively regulate grain size, probably via the same repressing pathway. GSK2 also interacts with MEI2-LIKE PROTEIN4 (OML4) encoded by the *LARGE1* gene. Plants with the loss-of-function mutation in *LARGE1* produce large and heavy grains. These data suggest that phosphorylation of OML4 by GSK2 enhances its stability and both proteins act on the same pathway to control grain size [126].

Because of the important role of *GSK2* in BR signaling, it is expected that its orthologues play a similar role in other cereal species. Indeed, the *GSK* genes were also identified in wheat and barley. Two glycogen synthase kinase 3/Shaggy-like kinase homologs were identified in wheat, and they are involved in the BR signaling pathway [127]. Moreover, the influence of wheat *GSK* on grain shape was confirmed in wheat lines derived from crossing *T. aestivum* with *T. sphaerococcum*, known as Indian dwarf wheat. *T. sphaerococcum* and possesses a gain-of-function allele *TaSG-D1*, which encodes a GSK homolog, and plants with this allele form short and round grains [128]. The negative role of *GSK* in grain size and weight were also confirmed in barley. The knockout of one *HvGSK* family member *HvGSK1.1* resulted in increased grain size and TGW [129].

With its ability to regulate several downstream genes, *GSK2* seems to be a crucial component in the BR signaling pathway, although its activity can also be modulated by other genes, such as *qGL3/OsPPKL1* and *GW5*. OsPPKL1 physically interacts with GSK2 and stabilizes its activity by dephosphorylation [130]. The *qgl3* mutants produce extra-large grains as compared to wild type, which confirms its suppressing function on BR signaling. 

As a positive grain size regulator, *GW5* acts in an opposite way to *qGL3*. *GW5* encodes a plasma-membrane localized protein with a calmodulin-binding domain, which can physically interact with GSK2 and suppress its kinase activity. GSK2 inactivation leads to the accumulation of non-phosphorylated forms of BZR1 and DLT, which enhances the expression of BR-responsive genes and BR-mediated seed growth [131]. 

*qGL3* and *GW5* modulate GSK2 kinase activity, indicating that both genes are key BR signaling regulators upstream of *GSK2*.

*GS5* and *GW10* are other important QTLs related to BR signaling identified in rice. A greater *GS5* expression is correlated with an increased grain width and weight. *GS5* encodes a serine carboxypeptidase, which can interact with the OsBAK1-7 extracellular leucine-rich repeat (LRR) domain, preventing it from undergoing endocytosis caused by OsMSBP1 [132,133,134]. In wheat, an association analysis revealed that *TaGS5-3A-T* expression was correlated with larger grains and greater TGW [135].

*GW10* encodes a cytochrome P450 subfamily member protein and has a positive influence on grain length. The exact function of *GW10* in the BR signaling pathway is unknown; however, its overexpression was associated with an upregulation of some BR-responsive genes [136]. Plant overexpressing *GW10* had longer grains, but the grain number per panicle was reduced as compared to wild type. Conversely, a *GW10* knockout exhibited a decreased grain length, but the grain number increased. 

#### 3.4.5. Ethylene

Ethylene was the first gaseous molecule discovered to function as a plant hormone. Its most known function is stimulation of fruit ripening, although ethylene regulates other growth and development processes, including cell division and expansion, tissue differentiation, seed germination, senescence, root hair formation, flowering, sex determination, and response to various stress factors [137,138]. 

The role of ethylene in grain size regulation is related to its ability to induce the cell division senescence process (Table 4). The positive or negative influence of ethylene on grain development also depends on the plant’s developmental stage. For instance, delay of leaf senescence before seed formation may increase the grain-filling rate and grain size. *OsSAMS1* gene expression is associated with ethylene production in leaves. The product of *OsSAMS1* is a target for OsFBK12, which is involved in the 26S proteasome pathway and targets the *OsSAMS1* product for degradation. Plants overexpressing *OsFBK12* produced reduced ethylene levels, showed delayed leaf senescence, and increased grain size [139]. However, ethylene production in early plant development stages may have a positive effect on grain yield. *OsPAO5* encodes POLYAMINE OXIDASE 5, which oxidizes polyamines and ethylene precursors [140]. *OsPAO5* knockout plants synthesized more ethylene, had longer mesocotyls, and showed increased grain weight and grain number. 

The positive effect of ethylene is also observed when it directly affects spikes during grain development. The knockout of a putative spermine synthase *OsSPMS1* resulted in elevated ethylene levels in developing seeds, increased cell expansion in hulls, and an increased grain size [141]. A similar positive effect was observed in the case of the ethylene responsive transcription factor ERF115. *OsERF115* expression is strongly induced by ethylene, and its overexpression significantly increases grain size and weight by promoting cell expansion and proliferation in spikelet hulls [142]. OsERF115 interacts with the transcriptional activator OsEIL1. The OsEIL1–OsERF115 module can directly or indirectly modulate the set of grain size related genes during spikelet growth and endosperm development. 

#### 3.4.6. Jasmonate

Jasmonic acid (JA) and its derivatives are plant hormones that mediate plant responses to biotic and abiotic stress factors and also regulate some developmental processes, including root formation, seed germination, leaf senescence, glandular trichome formation, and embryo and pollen development [143]. Recently the grain size regulatory function of two JA-associated genes in rice and wheat were reported (Table 4). 

The rice *TIFY* family genes act as repressors of JA action. They contain a characteristically conserved Jas domain, which is required for interaction with JA receptors [144]. Overexpression of a *TIFY* family member, *OsTIFY11b*, results in increased plant height and seed size. The enhanced growth of plants overexpressing *TIFY* genes might be associated with decreased JA sensitivity, which accelerates cell division in plant organs [144,145]. Contrary to this observation, studies in wheat showed that JA is required for enhanced grain yield. *tgw1* mutants exhibit reduced JA content and grain weight. The mutation causes the loss of function of a keto-acyl thiolase 2B (*KAT-2B*) gene involved in JA biosynthesis. *KAT-2B* overexpression increased the grain weight in transgenic plants [146].

**Table 4 ijms-24-16921-t004:** List of genes controlling grain size in rice, wheat, and barley involved in phytohormone signaling.

Gene or QTL Name	Gene ID	Protein Category	Type of Regulation	Regulated Process	Affected Trait	References
**Auxin**						
**Rice**						
Tillering And Small Grain 1 (TSG1)/FIB	Os01g0169800	Tryptophan aminotransferase-related protein 2	positive	cell proliferation/expansion	grain length	[72]
OsYUC11	Os12g0189500	Flavin-containing monooxygenase	positive	cell proliferation	grain size	[73]
Thousand-Grain Weight (TGW6)	Os06g0623700	IAA-glucose hydrolase	negative	cell expansion	grain size	[74]
Big Grain 1 (BG1)	Os03g0175800	plasma membrane-associated protein	positive	cell proliferation/expansion	grain size	[77,78]
OsARF4	Os01g0927600	Auxin response factor 4	negative	cell expansion	grain length	[82]
OsARF6	Os02g0164900	Auxin response factor 6	negative	cell expansion	grain length	[80]
OsARF12	Os04g0671900	Auxin response factor 12	positive	cell proliferation	grain size	[81]
OsAUX3	Os05g0447200	Auxin transporter-like protein 2	negative	cell expansion	grain length	[80]
qTGW3/OsGSK5	Os03g0841800	GSK3/SHAGGY-Like Kinase 41	negative	ns ^1^	grain length	[82]
Gnp4/LAX2	Os04g0396500	The RING-finger and wd40-associated ubiquitin-like protein	positive	cell expansion	grain length	[83]
OsIAA3	Os01g0231000	Auxin-responsive protein	negative	cell expansion	grain length	[83]
**Wheat**						
Thousand-Grain Weight (TaTGW6)	TraesCS6A02G526800LC	IAA-glucose hydrolase	negative	cell expansion	grain size	[75,76]
Big Grain 1 (BG1)	TraesCSU02G223800	plasma membrane-associated protein	positive	cell proliferation/expansion	grain size	[79]
**Cytokinins**						
**Rice**						
Big Grain3 (BG3)/OsPUP4	Os01g0680200	Purine permease	positive	cell proliferation/expansion	grain size	[87]
OsPUP7	Os05g0556800	Purine permease	positive	cell proliferation/expansion	grain size	[89,90]
AGO2	Os04g0615700	ARGONAUTE family protein	positive	cell proliferation/expansion	grain size	[88]
OsPIL15	Os01g0286100	bHLH transcription factor	negative	cell proliferation	grain size	[90]
OsSGL	Os02g0134200	Putative DUF1645 protein family member	positive	cell proliferation/expansion	grain size	[91]
qGL3 (OsPPKL1)	Os03g0646900	Serine/threonine-protein phosphatase	negative	ns	grain length	[92]
OsCKX2	Os01g0197700	Cytokinin dehydrogenase	negative	ns	grain size	[93]
**Wheat**						
TaCKX2.2	TraesCS3D02G143300	Cytokinin dehydrogenase	negative	ns	grain size	[94]
TaCKX6-D1	TraesCS3D02G143500	Cytokinin dehydrogenase	negative	ns	grain size	[95,96]
**Barley**						
HvCKX1		Cytokinin dehydrogenase	negative	ns	grain size	[97]
**Gibberellins**						
**Rice**						
BC12/GDD1	Os09g0114500	Kinesin-like protein	positive	cell expansion	grain size	[102]
GW6/OsGSR1	Os06g0266800	Gibberellic acid-stimulated transcript (GAST) family protein	positive	cell expansion	grain size	[103]
OsGARS9	Os07g0592000	Gibberellic acid-stimulated transcript (GAST) family protein	positive	cell expansion	grain size	[104]
GL10/OsMADS56	Os10g0536100	MADS-box transcription factor	positive	cell expansion	grain length	[105]
OsWRKY36	Os04g0545000	WRKY transcription factor	negative	cell proliferation/expansion	grain size	[106]
SLR1	Os03g0707600	DELLA protein	negative	cell proliferation/expansion	grain size	[106]
**Wheat**						
Rth-B1	TraesCS4B02G043100	DELLA protein	negative	ns	grain size	[55]
**Brassinosteroids**						
**Rice**						
DWARF 2 (D2)	Os01g0197100	Cytochrome P450 90D2	positive	cell expansion	grain size	[108,113]
DWARF 11 (D11)	Os04g0469800	Cytochrome P450 724B1	positive	cell expansion	grain size	[109,110,111]
BRD2	Os10g0397400	delta (24)-sterol reductase	positive	cell expansion	grain length	[112]
Slender Grain (SLG)	Os08g0562500	BAHD acyltransferase-like protein	positive	cell expansion	grain length	[113]
Brassinosteroid Insensitive1 (OsBRI1)	Os01g0718300	Leucin-rich-repeat receptor-like kinase	positive	cell expansion	grain size	[115]
OsBAK1	Os03g0440900	BRASSINOSTEROID INSENSITIVE 1-associated receptor kinase 1	positive	cell proliferation	grain length	[116]
XIAO	Os04g0576900	Leucine-rich repeat (LRR) receptor-like kinase	positive	cell proliferation	grain length	[147]
GSK3/Shaggy-like Kinase 2 (GSK2)	Os05g0207500	Shaggy-related protein kinase GSK2	negative	cell proliferation	grain size	[117]
Brassinazole Resistant 1 (BZR1)	Os07g0580500	Transcription factor OsBZR1	positive	cell expansion	grain length	[110]
Decreased Grain Size1 (DGS1)	Os03g0169800	transmembrane protein	positive	cell expansion	grain length	[118]
Dwarf And Low Tillering (DLT)	Os06g0127800	GRAS family protein 32	negative	cell proliferation	grain width	[117,119]
Dwarf And Low Tillering 2 (DLT2)	Os03g0723000	GRAS TF	negative	cell proliferation	grain width	[120]
Small Grain2 (SG2)	Os02g0450000	Ribonuclease H-like domain protein	positive	cell proliferation	grain size	[121]
Ovate Family Protein 1 (OsOFP1)	Os01g0226700	Transcription repressor	negative	ns	grain width	[124]
Ovate Family Protein 3 (OsOFP3)	Os01g0732300	Transcription repressor	negative	cell expansion	grain length	[125]
Ovate Family Protein 19 (OsOFP19)	Os05g0324600	Transcription repressor	negative	cell expansion	grain length	[121]
Large1	Os02g0517531	MEI2-LIKE PROTEIN4	negative	cell expansion	grain size	[126]
qGL3 (OsPPKL1)	Os03g0646900	Serine/threonine-protein phosphatase	negative	cell proliferation	grain length	[130]
GW5 (qSW5/GW5)	Os05g0187500	Calmodulin binding protein	positive	cell proliferation	grain size	[131]
Grain Size 5 (GS5)	Os05g0158500	Serine carboxypeptidase-like 26	positive	cell proliferation	grain width	[132,133]
GW10 (ORF1)	Os10g0515400	Cytochrome CYP89A2	positive	ns	grain length	[136]
**Wheat**						
TaD11	TraesCS2A02G331800	Cytochrome P450	positive	cell expansion	grain size	[114]
Grain Size 5 (TaGS5)	TraesCS6A02G220200	Serine carboxypeptidase-like 26	positive	ns	grain size	[135]
**Barley**						
GSK3/Shaggy-like Kinase1.1 (HvGSK1.1)	HORVU.MOREX.r3.3HG0252610	GSK3/SHAGGY-Like Kinase	negative	ns	grain size	[129]
**Ethylene**						
**Rice**						
OsFBK12	Os03g0171600	Kelch repeat-containing F-box family protein	positive	cell expansion	grain size	[139]
OsSPMS1	Os06g0528600	Aminopropyl transferase	negative	cell expansion	grain length	[141]
OsPAO5	Os04g0671300	Polyamine oxidase 5	negative	cell proliferation	grain length	[140]
OsERF115	Os08g0521600	Ethylene response factor	positive	cell proliferation	grain size	[142]
**Jasmonate**						
**Rice**						
OsTIFY11b	Os03g0181100	TIFY family protein	positive	ns	grain size	[144]
**Wheat**						
TGW1/KAT-2B	TraesCS6B01G432600	keto-acyl thiolase 2B	positive	cell expansion	grain size	[146]

^1^ Not specified

### 3.5. Transcriptional Regulation

Transcriptional regulators play crucial roles in the regulation of plant developmental processes. This large group includes transcription factors (TFs), long non-coding RNAs (lcRNAs), siRNAs, and chromatin modification regulators. TFs are sequence-specific DNA-binding proteins that recognize and bind to the regulatory cis-element in the target gene promoter region, and activate or suppress its transcription. lcRNAs can exert their regulatory function both in a cis and trans manner, and can also bind to other TFs, while siRNAs may be involved in DNA methylation. Numerous transcriptional regulators have been recognized as key elements of grain shape and weight regulation (Table 5).

#### 3.5.1. SPL Family Transcription Factors

SQUAMOSA promoter binding-like proteins (SPL) are a plant specific family of transcription factors. In rice, SPLs are involved in the differentiation between *indica*, *japonica*, and *javanica* (tropical japonica) varieties that also differ in grain shape [148,149]. OsSPL13 and OsSPL16 are encoded by two major rice grain size QTLs, *GLW7* (*GRAIN LENGTH AND WIEIGHT ON CHROMOSOME 7*) and *GW8* (*GRAIN WIDTH 8*), respectively. Sequence polymorphism in a tandem repeat motif in the *OsSPL13* 5′UTR alters its expression, and a greater expression is associated with larger grains in *javanica* rice [148]. *OsSPL13* promotes cell elongation in grain hull leading to the formation of significantly longer grains. A similar long grain phenotype is exhibited by *OsSPL16*; however, in this case, the hull growth is promoted by an increased cell proliferation [150]. Wheat orthologues of *GLW7* and *GW8* also encode SPL transcription factors that are involved in grain size regulation. *TaSPL13* harbors a miRNA recognition element (MRE) in the 3′UTR region and is a target for *miR156* [151]. The mutation in MRE induced by CRISPR/Cas9 led to a twofold increase in *TaSPL13* transcripts and was correlated with a decreased flowering time, tiller number, and plant height and, simultaneously, with increased grain size. TaSPL16 is also regulated by *miR156*, but its MRE is located in the last exon [152]. Similar to *TaSPL13*, *TaSPL16* expression has positive impacts on grain size [152,153].

Among other SPL TFs studied in rice, to date, *OsSPL12* [149] and *OsSPL18* [154] positively regulate grain size, which enhance cell elongation and cell proliferation in grain hulls, respectively.

#### 3.5.2. GRF Transcription Factors

Growth-regulating factors (GRFs) are specific TFs functionally categorized for growth regulation, which govern stem, leaf, and root development, flower morphogenesis, and seed development [155]. Similar to SPLs, several GRFs are regulated by miRNAs. 

In rice, three GRFs and one GRF-interacting factor (GIF) positively influence grain size and weight. *OsGRF1* and *OsGRF8* overexpression is positively correlated with grain size and weight, and both GRFs are negatively regulated by *miR396* [156]. The *miR396/GRF* module can further change miRNA expression, including miR408. These data suggest that the *miR396/GRF* module is part of a complex regulatory network based on miRNA/GRF interaction. OsGRF4 is encoded by QTL *GS2* (*GRAIN-LENGTH-ASSOCIATED 2*) and is also downregulated by miR396. In the rare *GS2* allele, the *miR396* binding site is disrupted, which results in greater *OsGRF4* expression. The *OsGRF4* expression enhances cell expansion and cell proliferation in spikelet hulls, leading to larger grain formation [157,158]. Moreover, *OsGRF4* can be upregulated by *GIF1*, which also has a positive impact on grain yield [158,159,160]. *GRF4* modulates grain yield by regulation of nitrogen and carbon metabolism [161]. This regulatory mechanism is based on the physical interaction of the GRF4 protein with the DELLA growth inhibitor in the homeostatic co-regulation of plant growth and carbon and nitrogen. GRF4 promotes nitrogen uptake, carbon assimilation, and growth, whereas DELLA inhibits these processes. GRF4 regulates the expression of multiple nitrogen metabolism genes, and its overexpression conferred both increased nitrogen uptake and assimilation. Increased GRF4 abundance was associated with increased grain number and grain weight in both rice and wheat [161,162]. In tetraploid durum wheat, the allelic variation in *GRF4* is also associated with grain weight. This association was confirmed by the introgression of a QTL fragment that carried *TtGRF4-A* from the wild emmer wheat into the durum wheat variety Svevo [163]. 

#### 3.5.3. bHLH Family Transcription Factors

The role of basic helix-loop-helix (bHLH) TFs in cell proliferation and differentiation means that they can also regulate cereal grain yield. bHLH are positive regulators of grain size by promoting cell expansion or cell proliferation in the spikelet hull. *An-1*, *PGL1* in rice and *TaPGS1* in wheat, regulate grain size independently from phytohormone signaling pathways [164,165,166], whereas other rice bHLHs, *BU1*, and *BUL1* are induced by brassinosteroids. *BU1* was identified in the BR-deficient mutant *brd1* after exogenous brassinolide treatment. *BU1* overexpression leads to an increased grain size [167]. *OsBUL1* knockout plants have erected leaves and smaller grains [168]. Jang and Li [169] identified a downstream gene, *OsBDG1,* upregulated by *OsBUL1,* whose overexpression had a positive effect on grain length. These data indicate that *BU1* and *OsBUL1* are transcriptional activators induced in response to BR.

#### 3.5.4. AP2/ERF Transcription Factors

AP2/ERF (APETALA2/ethylene-responsive element binding factors) are a large group of TFs involved in the regulation of various physiological processes, including morphogenesis, hormone signal transduction, response to stress factors, and metabolite regulation [170]. *SMOS1* encodes an unusual AP2 TF in rice with an imperfect AP2 domain. *SMOS1* contains an auxin response element in the promoter region, and its expression can be induced by auxin. A loss-of-function *smos1* mutant formed smaller organs, including grains, which may result from gene expression related to microtubule-based movement and DNA replication [84]. A direct target for SMOS1 is OsPHI-1, which is involved in cell expansion. These data indicate that SMOS1 is an auxin-dependent positive regulator of cell expansion in developing grains. 

*OsLG3* is a major grain length QTL in rice and a GWAS (genome-wide association study) showed that grain length alterations in the studied population were associated with a sequence polymorphism in the *OsLG3* promoter region [171]. A histological analysis showed that *OsLG3* positively regulates grain length by promoting cell proliferation. 

GWAS identified another grain size gene, *OsSNB*, from QTL *qGW7*. From identified haplotypes, Hap-3 has a 225 bp insertion in its promoter and is associated with the greatest grain width discovered in the japonica subspecies [172]. An *OsSNB* knockout significantly increased grain size, while its overexpression had the opposite effect, indicating that *OsSNB* is a negative grain size regulator. Histological analysis indicated that *OsSNB* affects cell expansion in glumes. 

#### 3.5.5. MADS-Box Transcription Factors

MADS-box transcription factors comprise one of the largest and best studied plant gene families. Initially, MADS-box genes were major regulators of floral organ development, although more recent studies revealed their function in the organogenesis of almost all organs and throughout all plant development stages [173].

*AFG1*(*Abnormal Flower* and *Grain1*) is one of the MADS-box transcription factors, which is involved in proper floret formation and grain development. Plants with the mutant allele *afg1* exhibit defects in floret anatomy, such as enlarged palea, elongated lodicules, and reduced stamen number [174]. Moreover, *afg1* mutant grains are markedly shorter, and the TGW is significantly reduced as compared to wild-type. *AFG1* acts as a transcriptional activator and may regulate the expression of several genes involved in cell proliferation and elongation during floret and grain development. 

#### 3.5.6. Epigenetic Regulation

Epigenetic modification refers to the changes in the chromatin structure or DNA modifications that are independent from the DNA sequence and can be inherited through mitotic or meiotic cell division. Epigenetic modification is responsible for the regulation of many developmental processes; however, little is known about epigenetic control of grain yield-related traits. *OsgHAT1* and *PANDA* genes are two examples of epigenetic control of grain size through histone modification.

*OsgIHAT1* encoded a GNAT-like protein with histone acetyltransferase activity and was identified in QTL *GW6a* (*Grain weight on chromosome 6*) in rice. OsgHAT1 protein is localized in the nucleus and functions as a transcriptional regulator. OsgHAT1 overexpression increases grain size via enhanced cell proliferation in spikelet hulls and accelerated grain filling. Moreover, its greater expression increases global histone H4 acetylation [175]. *PANDA* (*PANICLE NUMBER AND GRAIN SIZE*) is a Harbinger transposon-derived gene in rice, which epigenetically controls panicle number and grain size. A natural mutation in *PANDA* is associated with reduced panicle number and increased grain size [176]. The PANDA protein can bind to the core components of polycomb repressive complex 2 (PRC2), OsMSI1, and OsFIE2. PCR2 complex maintains gene repression through histone H3 methylation on Lys27 (H3K27me3). By interaction with PCR2, PANDA regulates H3K27me3 deposition in the target genes. Among these genes, both *OsMADS55* and *OsEMF1* are negative panicle number regulators but positive grain size regulators, which explains the role of PANDA in balancing between panicle number and grain size [176]. 

*RAV6* (*RELATED TO ABSCISIC ACID INSENSITIVE3 (ABI3)/VIVIPAROUS1 (VP1) 6*) is an example of an epiallele whose expression is controlled by the DNA methylation of its promoter. *RAV6* encodes a B3 DNA-binding protein, whose expression decreases grain size. The hypomethylation of *RAV6* promoter in *Epi-rav6* mutants causes an ectopic expression of the BR signaling genes *DWARF11* and *DWARF1*, which results in an increased leaf inclination and grain size [177].

MISSEN (MIS-SHAPEN ENDOSPERM) is the first lcRNA identified as an endosperm development regulator in rice. Moreover, its expression is regulated epigenetically. MISSEn is maternally expressed, and its expression is suppressed by histone H3 methylation (H3K27me3) after pollination. Zhou et al. [178] identified a T-DNA insertion mutant with the restored expression of MISSEN after pollination. Its expression caused severe abnormalities in the developing endosperm, leading to the formation of defective grains with bulges and dents. MISSEN interacts with the helicase family protein (HeFP) and suppresses its tubulin-related function during endosperm nucleus division and endosperm cellularization, resulting in abnormal cytoskeletal polymerization. 

#### 3.5.7. Other Transcriptional Regulators

GAGA-binding factors (GAFs) represent a small family of TFs in rice characterized by their specific binding property to the repeat motif (GA/TC) [179]. The function of GAFs for a long time remained unclear. Recent studies showed that two family members, *OsGBP1* and *OsGBP3*, regulate plant growth and grain development. *OsGBP1* expression has a positive impact on grain length, whereas *OsGBP3* has the opposite effect. Interestingly, both genes function synergistically in the regulation of plant growth and grain width. The RNAi-based silencing of both *OsGBP1* and *OsGBP3* resulted in semi-dwarf plants with a decreased grain width [179]. These data suggest that functional divergence of GAFs is tissue specific.

*KNOX* (*Knotted1-like homeobox*) genes encode homeodomain-containing TFs; however, little is known about their function in plant development. Genomic studies showed that one *KNOX* family member, *HOS59,* is a negative rice grain size regulator [180]. HOS59 acts as a transcriptional repressor of several downstream genes that positively regulate grain size, such as *OsPL13*, *OsPL18*, and *PGL2*. 

*GL6* is a rice grain length QTL, which encodes the PLATZ transcription factor (plant AT-rich sequence- and zinc-binding protein). *GL6* has a positive impact on grain length by promoting cell proliferation in young panicles and grains, although it negatively affects grain number per panicle [181]. 

*OsNF-YC10* is a member of the *NF-Y* transcription factors family, which encodes a putative histone transcription factor. Rice lines with an *osnf-yx10* knockout produced thin, narrow, and light grains [182]. Cytological studies showed that decreased width resulted from reduced cell number in spikelet hulls, especially in the grain-width direction. Further analysis revealed that *OsNF-YC10* influences the expression of other grain size-related genes, like *GW7* and *GW8,* and cell cycle-related genes, such as *CYCD4*, *CYCA2.1*, *CYCB2.1*, *CYCB2.2*, and *E2F2* [182]. 

*TH1* (*TRIANGULAR HULL1*)/*AFD1* (*ABNORMAL FLOWER AND DWARF1*) rice mutants exhibit squeezed, thin grains, which results from reduced spikelet width [183,184]. *TH1* encodes an ALOG family protein, which localizes in the nucleus and possesses transcriptional repression activity. A microscopic analysis indicated that a reduced spikelet width resulted from a decreased cell expansion [184]. 

*OsFD2* encodes a basic leucine zipper (bZIP) transcription factor, which negatively regulates rice grain size. *OsFD2* overexpression reduces the cell number and cell elongation in spikelet hulls. OsFD2 could be an element of florigen repression complex (FRC), as it interacts with the TERMINAL FLOWER1 homolog OsCEN2. *OsCEN2* encodes phosphatidylethanolamine-binding protein, which mediates the transition from the vegetative to the reproductive stage [185].

**Table 5 ijms-24-16921-t005:** List of genes controlling grain size in rice and wheat, involved in transcriptional regulation.

Gene or QTL Name	Gene ID	Protein Category	Type of Regulation	Regulated Process	Affected Trait	References
**Rice**						
GLW7/OsSPL13	Os07g0505200	Squamosa promoter-binding-like protein	positive	cell expansion	grain size	[148]
GW8/OsSPL16	Os08g0531600	Squamosa promoter-binding-like protein	positive	cell proliferation	grain size	[150]
OsSPL12	Os06g0703500	Squamosa promoter-binding-like protein	positive	cell expansion	grain width	[149]
OsSPL18	Os09g0507100	Squamosa promoter-binding-like protein	positive	cell proliferation	grain width	[154]
OsGRF1	Os02g0776900	Growth-regulating factor 1	positive	ns ^1^	grain size	[156]
GS2/OsGRF4	Os02g0701300	Growth-regulating factor 4	positive	cell proliferation/expansion	grain size	[157,158,159]
OsGRF8	Os11g0551900	Growth-regulating factor 8	positive	ns	grain length	[156]
OsGIF1	Os03g0733600	GRF-interacting factor 1	positive	cell expansion	grain size	[158,159,160]
An-1	Os04g0350700	bHLH transcription factor	positive	cell expansion	grain length	[165]
PGL1	Os03g0171300	bHLH transcription factor	positive	cell expansion	grain length	[164]
OsBUL1	Os02g0747900	bHLH transcription factor	positive	cell expansion	grain size	[168,169]
Brassinosteroid Upregulated 1 (BU1)	Os06g0226500	bHLH transcription factor	positive	ns	grain size	[167]
SMOS1	Os05g0389000	AP2-like ethylene-responsive transcription factor	positive	cell expansion	grain size	[84]
OsLG3	Os03g0183000	AP2 domain class transcription factor	positive	cell expansion	grain size	[171]
OsSNB	Os07g0235800	AP2 transcription factor	negative	cell expansion	grain size	[172]
AFG1	Os02g0682200	MADS-box transcription factor 6	positive	cell proliferation/expansion	grain size	[174]
Grain Weight 6a (GW6a)/OsgHAT1	Os06g0650300	GNAT-like histone acetyltransferase 1	positive	cell expansion	grain length	[175]
PANDA	Os07g0175100	Harbinger transposon derived protein	negative	ns	grain size	[176]
RAV6	Os02g0683500	B3 DNA-binding domain-containing protein	negative	ns	grain size	[177]
MISSEN	XLOC_057324	Long noncoding RNA	negative	cell proliferation	grain size	[178]
OsGBP1	Os06g0130600	GAGA-binding factor	positive	ns	grain length	[179]
OsGBP3	Os10g0115200	GAGA-binding factor	negative	ns	grain length	[179]
HOS59	Os06g0646600	KNOX II transcription factor	negative	cell expansion	grain length	[180]
GL6	Os06g0666100	plant AT-rich sequence- and zinc-binding (PLATZ) protein	positive	cell proliferation	grain length	[181]
OsNF-YC10	Os01g0346900	NF-Y transcription factor	positive	cell proliferation	grain width	[182]
TH1/AFD1	Os02g0811000	ALOG domain associated protein	positive	cell expansion	grain size	[183,184]
OsFD2	Os06g0720900	basic leucine zipper (bZIP) transcription factor	negative	cell proliferation/expansion	grain size	[185]
**Wheat**						
GLW7/TaSPL13	TraesCS2A02G232400	Squamosa promoter-binding-like protein	positive	cell expansion	grain size	[151]
GW8/OsSPL16	TraesCS7A02G260500	Squamosa promoter-binding-like protein	positive	cell proliferation	grain size	[152,153]
TaPGS1	TraesCS1D02G094000	bHLH transcription factor	positive	ns	grain size	[166]
TaGRF4-A1/TaGRF3-2A	TraesCS2A02G435100	Growth-regulating factor	Positive	ns	grain size	[161,162,163]

^1^ Not specified.

### 3.6. miRNAs

miRNAs are short, noncoding RNA molecules that post-transcriptionally regulate target gene expression by either mRNA degradation or translation blockade. Several miRNAs are involved in controlling grain size and weight by regulating the expression of genes that are positive or negative grain development regulators (Table 6). In the phytohormone signaling Section 3.4, we described the regulatory function of the *miR167a-OsARF6-OsAUX3* module and *miR167-OsARF12* module on grain size in response to auxin. The interaction of *miR156* with SPL TFs, and *miR396* and *miR408* with GRF TFs in grain size regulation was mentioned in the transcriptional regulation Section 3.5. There are also a few miRNAs that regulate grain size and weight via different pathways than those previously described.

*miR397* is a positive grain size regulator in rice. The overexpression of two isoforms of *miR397*, *miR397a*, and *miR397b* promotes panicle branching and increases grain size leading to up to a 25% greater grain yield [186]. *miR397* downregulates the expression of its target *OsLAC*, which encodes a laccase-like protein. Since plants overexpressing *OsLAC* phenotypically resembled BR-deficient mutants, and many BR-related genes were differentially expressed, it is hypothesized that *OsLAC* is involved in BRs sensitivity control [186].

Contrary to *miR397*, *miR530* acts as a negative grain size regulator in rice. A direct target for *miR530* is a PULS3 domain containing protein-encoding gene *OsPL3*. *OsPL3* positively regulates grain yield by promoting panicle branching, cell expansion, and cell proliferation in spikelet hulls [187]. Additionally, *miR530* is activated by OsPIL15, a rice homolog of phytohormone interacting factors (PIFs). The role of PIFs in hormone signaling has been well-studied in *Arabidopsis*; however, little is known about their function in cereals. The above data indicate that *OsPILs* (phytochrome-interacting factor-like) genes may regulate grain yield in rice by interaction with miRNAs.

**Table 6 ijms-24-16921-t006:** List of miRNAs controlling grain size in rice.

Gene or QTL Name	Gene ID	Protein Category	Type of Regulation	Regulated Process	Affected Trait	References
**Rice**						
miR167a	MI0000676	miRNA	positive	cell expansion	grain length	[80]
miR156k	MI0001090	miRNA	negative	cell proliferation	grain size	[154]
miR397a	MI0001049	miRNA	positive	ns ^1^	grain size	[186]
miR397b	MI0001050	miRNA	positive	ns	grain size	[186]
miR530	MI0003203	miRNA	negative	cell proliferation/expansion	grain size	[187]

^1^ Not specified.

### 3.7. Grain Filling and Sugar Transport Associated Regulators

Final grain size and weight is also determined by endosperm development regardless of the hull growth. The grain-filling stage is a critical period for endosperm development and storage material accumulation. The genes controlling nutrient transport and starch synthesis (Table 7) are important during this stage, and any disturbances in their function may lead to the production of smaller, less filled grains.

Efficient sugar efflux is crucial in seed development and is maintained by membrane-localized transporters. Recently, a new type of sugar transporter named SWEET (Sugars Will Eventually be Exported Transporters) was identified in Arabidopsis [188]. They represent a highly conserved gene family and are ubiquitous among the plant kingdoms. For example, there are 21 *SWEET* genes in rice, 108 in wheat [189], and 23 in barley [190]. SWEETs are bidirectional sugar transporters capable of both uptake and efflux and are involved in phloem loading, pollen viability, grain filling, and plant-pathogen interactions [189]. In rice, the function of *SWEET4*, *SWEET11*, and *SWEET15* genes were experimentally validated [191,192,193]. They exhibit strong expression in developing grains between 1 and 14 DAP and are mostly expressed in the nucellus, nucellar projection, and endosperm transfer cells. All three transporters are crucial for proper grain filling, as their knockout results in the defective endosperm, which is manifested as concavities in the mature caryopses or the formation of very thin grains. SWEET transporters are also crucial for barley grain development. Recent studies revealed a key role for *HvSWEET11b* in grain filling and endosperm formation [190]. The greatest *HvSWEET11b* transcript level is observed between 6 and 18 days after flowering, and its tissue localization is similar to the rice SWEET transporters mentioned above. *HvSWEET11b* knockout results in the plant’s inability to form viable grains, while the partial repression of its transcription leads to decreased endosperm cell number, reduced starch and protein accumulation, and grain size reduction. Moreover, *HvSWEET11b* is capable of cytokinin transport, which has an additional enhancing influence on endosperm development.

SUTs (sucrose transporters) are another class of plant sugar transporters. Rice OsSUT2 is localized in the tonoplast [194]. A histological analysis indicated that *OsSUT2* is greatly expressed in the pedicels of fertilized spikelets, and seed coat cross cell layers, among other tissues. OsSUT2 functions as a symport transporter in sucrose uptake from vacuoles. The *ossut2* loss-of-function mutants exhibit a reduced growth, and a significant reduction in tiller number, plant height, TGW, and root dry weight, which indicates OsSUT3 is a key sucrose transporter, not only during grain development, but during the whole plant’s growth period.

Apart from sugar transporters, other genes are also involved in grain filling. Rice *GIF1* (*GRAIN INCOMPLETE FILLING1*) encodes cell wall invertase, which is required for carbon partitioning during early grain filling [195]. *GIF1* overexpression increases the grain-filling rate, which has a positive influence on the final grain yield. Sequence divergence in the *GIF1* promoter region between cultivated and wild rice may suggest that *GIF1* is a domestication signature. 

*GRF1* (*GRAIN-FILLING RATE1*) is a constitutively expressed, membrane-localized protein in rice. The favorable effect of *GRF1* on grain filling was determined in an association study with 117 rice accessions [196]. Further analysis of the *GRF1^Ludado^* accession (possessing an allele that promoted grain-filling rate) revealed that GRF1 can interact with the Rubisco small units and possibly increase its activity. Moreover, the expression of the cell wall invertase gene *OsCIN1* was also greater in plants harboring *GRF1^Ludado^*, which promoted sucrose unloading during the grain-filling stage. 

*OsPK3* is a pyruvate kinase encoding gene and is a positive grain-filling regulator in rice [197]. *OsPK3* is constitutively expressed, but the greatest expression was observed in the leaf and developing caryopsis. The localization in tissues was involved in sucrose transport and unloading, such as nucellar projection, nucellar epidermis, aleurone layer cells, and ovular vascular trace in the caryopsis supports the evidence of its biological function in grain filling by sugar transport regulation. Subcellular localization revealed that OsPK3 is associated with mitochondria. *OsPK3* disruption in the T-DNA insertion mutant caused pleiotropic defects, such as decreased plant height and significantly reduced TGW. 

Grain filling in rice is also regulated by ABA in a temperature-dependent manner. MATE-transporter, a product of the *DG1* (Defective grain-filling 1) gene is involved in long-distance ABA transport from leaves to developing seeds [198]. The *dg1* mutants produce unfilled grains with a significantly reduced TGW. At temperatures above 30 °C, *DG1* expression increased two-fold in WT plants, which was associated with an increased ABA transport from nodes to caryopses, while in the *dg1* mutants this transport was suppressed. 

**Table 7 ijms-24-16921-t007:** List of genes controlling grain size in rice and barley involved in sugar transport and grain filing.

Gene or QTL Name	Gene ID	Protein Category	Type of Regulation	Regulated Process	Affected Trait	References
**Rice**						
OsSWEET4	Os02g0301100	Bidirectional sugar transporter	positive	grain filling	grain size	[191]
OsSWEET11	Os08g0535200	Bidirectional sugar transporter	positive	grain filling	grain size	[192]
OsSWEET15	Os02g0513100	Bidirectional sugar transporter	positive	grain filling	grain size	[193]
OsSUT2	Os12g0641400	Sucrose transporter	positive	grain filling	grain size	[194]
Grain-Filling Rate1 (GRF1)	Os10g0508100	membrane-localized protein	positive	grain filling	grain size	[196]
OsPK3	Os04g0677500	Pyruvate kinase	positive	grain filling	grain size	[197]
DG1	Os03g0229500	MATE transporter	positive	grain filling	grain size	[198]
**Barley**						
HvSWEET11b	HORVU.MOREX.r3.7HG0684580	Bidirectional sugar transporter	positive	grain filling	grain size	[190]

### 3.8. Other Regulators

Genes assigned to this group have distinct molecular functions, other than those described in previous sections (Table 8).

Cytochrome P450 (CYP) is one of the largest plant protein families that is involved in almost all physiological processes, and some members are important in grain development regulation. For example, CYP704A3 has a negative effect on rice grain length [199]. CYP704A3 is a target for *osamiRf10422-akr*, although whether and how this miRNA controls CYP704A3 expression requires further investigation. Another rice CYP, CYP78A13 encoded by *GE* (*GIANT EMBRYO*), controls the balance between embryo and endosperm size. *GE* is expressed mostly in the interface between the embryo and endosperm tissues. GE overexpression results in a small embryo and an enlarged endosperm, whereas the loss-of-function mutation leads to the opposite effect [200]. Variations in the *bg2* mutant coding sequence are associated with an increased grain length [201]. In a wheat P450 family member, *TaCYP78A3* is expressed in reproductive organs, and its expression is positively correlated with grain size. *TaCYP78A3* silencing reduced the cell number in seed coats, resulting in an 11% decrease in grain size [202].

Grain length QTL in rice, which has been assigned different names (*GW7, GL7, or SLG7*) encodes the TONNEAU1-recruting motif protein. This protein positively affects the grain length by promoting cell division in the longitudinal direction and decreasing in the traverse direction [203,204,205]. The same grain length regulation mechanism was observed in the wheat ortholog *TaGW7*, which is present in the B and D genomes [206].

Kinesin 13A, which possesses microtubule depolymerization activity, promotes cell elongation in glumes, resulting in an increased final rice grain length [207,208]. It is speculated that kinesin 13A changes the orientation of microtubules, which may facilitate vesicle transport from the Golgi apparatus to the cell surface. This, in turn, may affect cellulose microfibril orientation and cell elongation [208]. OsIQD14 is another microtubule-associated protein, which controls grain shape. *OsIQD14* is auxin-inducible and is highly expressed in grain hull cells. Plants deficient in OsIQD14 produce short and wide seeds with an increased overall grain yield, while OsIQD14 overexpression results in narrow, long seeds, caused by changed microtubule alignment [209].

ABC1-like kinase is a newly found atypical plant kinase, which is localized in chloroplasts. ABC1 is encoded by the *OsAGSW1* gene, whose expression is observed in vascular bundles in shoot, hull, and caryopsis. *OsGSW1* overexpression increased cell number in external parenchyma cells in hulls, which resulted in a significantly increased grain size, grain-filling rate, and TGW [210].

*FUWA* is an evolutionarily conserved gene that controls multiple rice agronomic traits, such as panicle architecture, grain shape, and grain weight. *FUWA* encodes NHL repeat-containing protein and is preferentially expressed in the root meristem, shoot apical meristem, and inflorescences. *FUWA* negatively regulates grain size by suppressing the expression of cell cycle genes. The loss-of-function mutant *fuwa* produced wider and thicker grains with up to 16.7% greater TGW [211].

*TGW2* is a recently discovered semidominant QTL for grain width and weight in rice. *TGW2* encodes the plasma membrane-localized protein OsCNR1 (CELL NUMBER REGULATOR 1). *TGW2* is expressed in young panicles, and it suppresses cell proliferation and expansion in glumes by interaction with KRP1, a plant cell cycle regulator. A SNP variant 1818 bp upstream of *TGW2* is responsible for its reduced expression, which results in an increased grain width and weight [212]. 

Melatonin is considered another plant growth regulator with functions similar to auxin; however, its role in grain yield regulation remained unexplored until recently. The *OsCOMT* gene in rice encodes caffeic acid O-methyltransferse, which is involved in melatonin biosynthesis. Genomic studies showed that *OsCOMT* significantly delays leaf senescence at the grain-filling stage by restraining the degradation of chlorophyll and chloroplasts, which improves photosynthetic activity and availability of assimilates [213]. Moreover, *OsCOMT* overexpression significantly increased the number and size of vascular bundles in culms and leaves. Plants overexpressing *OsCOMT* showed a significant increase in grain size, grain number per plant, and TGW. This positive effect may result from the crosstalk between melatonin and cytokinin. Exogenous melatonin application changed the gene expression related to cytokinin metabolism, i.e., upregulated genes responsible for cytokinin biosynthesis and downregulated genes responsible for their degradation [213]. 

**Table 8 ijms-24-16921-t008:** List of other genes associated with grain size in rice and wheat.

Gene or QTL Name	Gene ID	Protein Category	Type of Regulation	Regulated Process	Affected Trait	References
**Rice**						
OsCYP704A3	Os04g0573900	Cytochrome CYP704A3	negative	ns ^1^	grain length	[199]
GIANT EMBRYO/BG2	Os07g0603700	Cytochrome CYP78A13	negative	embryo development	grain length	[200,201]
Grain Length 7 (GL7/GW7/SLG7)	Os07g0603300	TONNEAU1-recruiting motif protein	positive	cell proliferation	grain length	[203,204,205]
OsKinesin-13A/SRS3	Os05g0154700	Kinesin 13A	positive	cell expansion	grain length	[207,208]
OsIQD14	Os08g0115200	microtubule-associated protein	positive	cell expansion	grain length	[209]
OsAGSW1	Os05g0323800	ABC1-like kinase	positive	cell proliferation	grain size	[210]
FUWA	Os02g0234200	NHL repeat-containing protein	negative	proliferation	grain width	[211]
TGW2	Os02g0763000	CELL NUMBER REGULATOR 1 (OsCNR1)		cell proliferation	grain size	[212]
OsCOMT	Os08g0157500	Flavone 3’-O-methyltransferase 1	positive	ns	grain size	[213]
**Wheat**						
TaCYP78A3	KP768392	Cytochrome CYP78A3	positive	ns	grain size	[202]
TaGW7	TraesCS2A01G176000	TONNEAU1-recruiting motif protein	positive	cell proliferation	grain length	[206]

^1^ Not specified.

## 4. Future Perspectives

In the past decade, significant progress has been made in grain yield control research. Many grain size regulators were identified in rice; however, the understanding of their function at the molecular level and the relationship with other genes and transcriptional regulators is still insufficient. Moreover, the knowledge gained in rice studies must be applied to a greater extent to the research of wheat, barley, and other important cereal crops. Research of important agronomic traits in oat, rye, and triticale is often neglected because of the lesser economic importance of these crops. There is also a large gap between the recent discoveries of the genetic control of grain size and the implementation of this knowledge in plant breeding. In this situation, the limiting factor is a trade-off between grain yield and other important agronomic traits that are often negatively correlated. A major challenge in the development of new varieties with improved grain yield is to find the optimal balance between grain size and grain quality, disease resistance, and tolerance to abiotic stress factors. Therefore, further studies are needed to evaluate the impact of desired grain size alleles on the nutritional value and flour properties of wheat and rice and on malting properties in barley. Recent advances in genetic engineering may help to overcome a large part of these challenges. New plant breeding techniques that use genome editing allow for the generation of both loss-of-function or gain-of-function mutants in a single generation. This technique can be used to directly introduce the desired trait into elite cultivars without changing its genetic background. This approach eliminates the need for laborious and time-consuming backcrosses that are performed in conventional breeding.

In conclusion, the latest discoveries of genetic regulation of grain size in cereals have great application potential in breeding new cultivars that will meet the increasing food demand and simultaneously fulfill sustainable agriculture requirements.

## Figures and Tables

**Figure 1 ijms-24-16921-f001:**
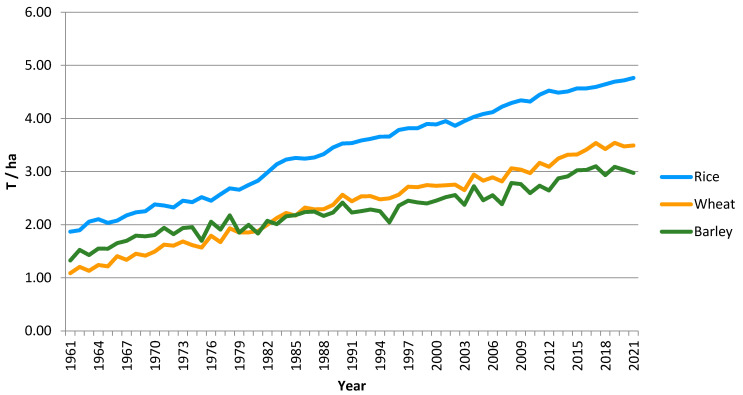
Global yield (tons per hectare) of wheat, barley, and rice from 1961 to 2021 according to FAOSTAT (2021, accessed on 1 October 2023 www.fao.org/faostat).

**Figure 2 ijms-24-16921-f002:**
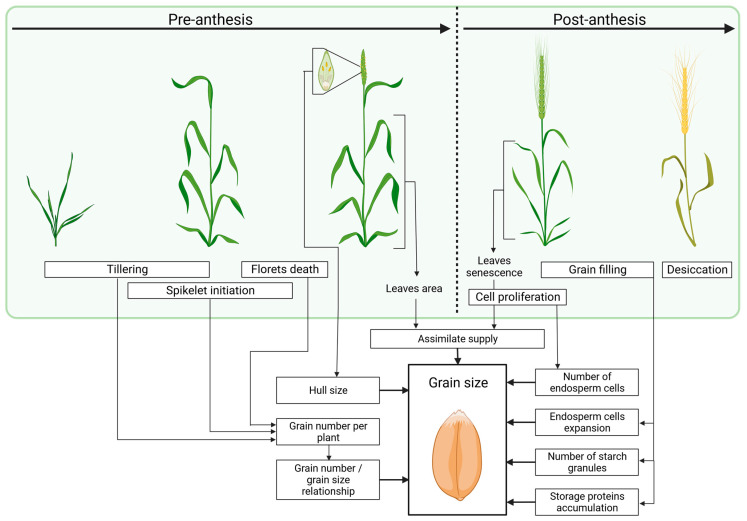
Major factors that influence grain size during the pre-anthesis and post-anthesis developmental stages; adapted from “IconPack—Wheat”, by BioRender.com (2023). Retrieved from https://app.biorender.com/biorender-templates, accessed on 1 November 2023.

## Data Availability

All data are included within the article or its Appendix A files.

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
