# Peer review of "Genetic Basis of Grain Size and Weight in Rice, Wheat, and Barley"

_ijms, 2023, doi:10.3390/ijms242316921_

Round 1

Reviewer 1 Report

Comments and Suggestions for Authors

The proposed review article summarizes relevant data on the genetic background underlying such important yield components as grain size and weight in the most important cereal crops, rice, wheat, and barley. Factors determining grain size and weight during plant development are discussed, and the latest discoveries in the field of genetic and molecular mechanisms of grain size such as G protein system, ubiquitin proteasome system, phytohormone signaling, transcriptional regulation, and other physiological processes are highlighted.

However, I have some comments and suggestions:

 1)      Can rice, wheat and barley be considered closely related cereals as stated in the manuscript (lines 18 and 19; 83 and 84)? Barley and wheat have homeologous genomes, while rice is closer phylogenetically to corn and sorghum. The two branches diverged more than 35 million years ago. I recommend providing data on the number of chromosomes (linkage groups) and the degree of relatedness of the genomes of all three species. Please correct.

2)      I also recommend to give information on the chromosome location (linkage groups) for at least some genes considered.

Formal notes:

-          Please give latine names for the species considered.

-          Please edit the names of tables 2, 3, 5, 6, 7, 8 in accordance with content. For example, Table 2 named “List of genes controlling grain size in rice, wheat, and barley involved in the ubiquitin proteasome system” contains information only on the rice and wheat genes and etc.

-          Line 311 – Please replace MPAK by MAPK (line 319), OsMPAK6 by OsMAPK (line 326).

-          Line 430 - Please correct: cytokines

-          Lines 700-702 Please correct the phrase: “From identified haplotypes, Hap-3 has a 225 bp insertion in its promoter and has the greatest grain width  ….” Please replace by: “From identified haplotypes, Hap-3 has a 225 bp insertion in its promoter and is associated with the greatest grain width ….”

-          Line 898 Please change number of Table 13 by Table 8.

Reviewer 2 Report

Comments and Suggestions for Authors

Dear Authors,

I appreciate your effort concerning the genetics of grain size among rice, wheat, and maize.

You started to say about green revolution genes, but subsequently started with another sentence like these varieties. No connection exists between the sentences. Please check inside your manuscript for the comment.

Those new varieties bearing these genes have been producing... please write with connectivity to the previous sentence, since the later sentence is expected to be connected to the previous one.

Perhaps you need to alter the sentence.

Since it is presumed that those varieties were already developed and being used for producing advanced lines)

But I suggest to provide more scientific inputs in writing: 

When you discuss the seed size and numbers according to the evolutionary and genetic basis, you also have to give attention to metabolic costs. Accordingly,  over generations, evolutionary adaptation would occur due to the selection pressure on specific genes to coordinate a complex trait.

The seed size is inversely related to seed numbers in any crop or plant species due to metabolic cost; please rewrite slightly according to the thermodynamics point of view. Hence, the readers can understand easily.

An inherent and appropriate management of metabolic cost to size and numbers leads to evolutionary and environmental adaptations over generalizations.

Please check it may be useful: https://nph.onlinelibrary.wiley.com/doi/pdf/10.1111/j.1469-8137.2009.02878.x

A review needs to be very precise and written in a scientific way.

For instance, you have mentioned "Although the grain size range is relatively limited, it still provides the potential for substantial grain yield improvement. For instance, in domesticated rice the grain size ranges from 3 mm to 11 mm in length and from 1.2 71 mm to 3.8 mm in width.

Could you please rewrite as follows or as you wish as an example here?

The existence of a relative range of 3 to 11 mm in length and 1.2 to 3.8 mm in width among domesticated rice varieties would provide a clue for substantial grain yield.

Middle content of introduction missing references for CRISPR-CAS9 etc. At least cite one reference when you strongly emphasize a point.

How you can write like this:

Growth-regulating factors (GRFs) are plant specific transcription factors whose primary function is regulation of stem, leaf, and root development, and flower and seed formation.

Everyone knows that it must be a plant-specific transcription factor only when you are talking about plants.

You could write:

Specific TFs functionally categorized for growth regulation govern the stem, leaf, and root and flower morphogenesis and seed development.

Your manuscript provides good scientific content but lacks precise scientific writing and presentation; hence, it does not enable the readers to understand easily. Writing precisely, scientifically, is more important than vastly.

When you mention annotated candidate genes, you must mention them as candidate genes rather than simply as genes.

I suggest taking your time to thoroughly revise the article and resubmit it again according to the editor's decision.

Comments on the Quality of English Language

Reviewer 3 Report

Comments and Suggestions for Authors

The manuscript "Genetic bases of grain size and weight in rice, wheat and barley" by Gasparis and Miłoszewski is a brilliant and comprehensive review dedicated to the relevant problem of genetic and hormonal determination of grain size in the main cereals: wheat, barley, and rice. The authors have done an excellent job summarizing multiple studies. The material is clearly presented and systematized, so it can be recommended even for students as essential reading to understand the fundamentals of grain shape and size determination.

I have only a few comments on the manuscript.

1. Please check if all gene names are in italics, for example, in lines 39 (sd1), 40 (sdw1), 111 (Vrs1, INT-C), 214 (OsMADS1), and 599 (OsERF115). It is important to differentiate whether the gene or protein is described.

2. Figure 2. Please redesign the scheme so that the lines from the "Tillering" and "Florets death" boxes do not intersect with the "Spikelet inititation" box. In its current form, it is confusing and appears that these boxes interact with each other.

3. In 3.5.2 GRF transcription factors only rice GRFs and miR396s are considered. We recommend using the following references to enhance this section, as they describe the effects of GRF in durum and common wheat:

1) Li, S., Tian, Y., Wu, K. et al. Modulating plant growth–metabolism coordination for sustainable agriculture. Nature 560, 595–600 (2018).

2) Avni, R., Oren, L., Shabtay, G., Assili, S., Pozniak, C., Hale, I., ... & Distelfeld, A. (2018). Genome based meta-QTL analysis of grain weight in tetraploid wheat identifies rare alleles of GRF4 associated with larger grains. Genes, 9(12), 636.

3) Bazhenov, M. S., Chernook, A. G., Bespalova, L. A., Gritsay, T. I., Polevikova, N. A., Karlov, G. I., ... & Divashuk, M. G. (2021). Alleles of the GRF3-2A gene in wheat and their agronomic value. International Journal of Molecular Sciences, 22(22), 12376.

4) Zan, T., Zhang, L., Xie, T., & Li, L. (2020). Genome-wide identification and analysis of the growth-regulating factor (GRF) gene family and GRF-interacting factor family in Triticum aestivum L. Biochemical genetics, 58, 705-724.

5) Kroupin, P. Y., Chernook, A. G., Bazhenov, M. S., Karlov, G. I., Goncharov, N. P., Chikida, N. N., & Divashuk, M. G. (2020). Allele mining of TaGRF-2D gene 5'-UTR in Triticum aestivum and Aegilops tauschii genotypes. PLoS One, 15(4), e0231704.

I wish the authors great success in their research activities and new discoveries in plant science.

Kind regards,

Reviewer.

Round 2

Reviewer 2 Report

Comments and Suggestions for Authors

Dear Authors,

Revisions are acceptable. However, please check thoroughly once for typos or any.

If you wish, please change the title; in itself, there are some punctuation errors.

The title could be: 

Genetic basis and comparison of grain size and weight among rice, wheat, and barley.

Author Response

Thank you once again for your comments. We slightly changed the title. The manuscript have been submitted to the professional English editor for proofreading to correct all typos and punctation.